# Emerging Evidence of Pathological Roles of Very-Low-Density Lipoprotein (VLDL)

**DOI:** 10.3390/ijms23084300

**Published:** 2022-04-13

**Authors:** Jih-Kai Huang, Hsiang-Chun Lee

**Affiliations:** 1Department of General Medicine, Kaohsiung Medical University Hospital, Kaohsiung Medical University, Kaohsiung 80708, Taiwan; eric86425@gmail.com; 2Division of Cardiology, Department of Internal Medicine, Kaohsiung Medical University Hospital, Kaohsiung Medical University, Kaohsiung 80708, Taiwan; 3Department of Internal Medicine, School of Medicine, College of Medicine, Kaohsiung Medical University, Kaohsiung 80708, Taiwan; 4Lipid Science and Aging Research Center, College of Medicine, Kaohsiung Medical University, Kaohsiung 80708, Taiwan; 5Institute of Medical Science and Technology, National Sun Yat-sen University, Kaohsiung 80708, Taiwan; 6Graduate Institute of Animal Vaccine Technology, National Pingtung University of Science and Technology, Pingtung 91201, Taiwan

**Keywords:** very-low-density lipoprotein (VLDL), VLDL receptor (VLDLR), metabolic syndrome (MetS), insulin resistance, metabolic associated fatty liver disease (MAFLD), endocrinological disorders, cardiovascular disorders, cognitive dysfunction, cancer, atrial myopathy

## Abstract

Embraced with apolipoproteins (Apo) B and Apo E, triglyceride-enriched very-low-density lipoprotein (VLDL) is secreted by the liver into circulation, mainly during post-meal hours. Here, we present a brief review of the physiological role of VLDL and a systemic review of the emerging evidence supporting its pathological roles. VLDL promotes atherosclerosis in metabolic syndrome (MetS). VLDL isolated from subjects with MetS exhibits cytotoxicity to atrial myocytes, induces atrial myopathy, and promotes vulnerability to atrial fibrillation. VLDL levels are affected by a number of endocrinological disorders and can be increased by therapeutic supplementation with cortisol, growth hormone, progesterone, and estrogen. VLDL promotes aldosterone secretion, which contributes to hypertension. VLDL induces neuroinflammation, leading to cognitive dysfunction. VLDL levels are also correlated with chronic kidney disease, autoimmune disorders, and some dermatological diseases. The extra-hepatic secretion of VLDL derived from intestinal dysbiosis is suggested to be harmful. Emerging evidence suggests disturbed VLDL metabolism in sleep disorders and in cancer development and progression. In addition to VLDL, the VLDL receptor (VLDLR) may affect both VLDL metabolism and carcinogenesis. Overall, emerging evidence supports the pathological roles of VLDL in multi-organ diseases. To better understand the fundamental mechanisms of how VLDL promotes disease development, elucidation of the quality control of VLDL and of the regulation and signaling of VLDLR should be indispensable. With this, successful VLDL-targeted therapies can be discovered in the future.

## 1. Introduction of Very-Low-Density Lipoprotein (VLDL)

### 1.1. Structure Characteristics of VLDL

The major lipid content transported by the VLDL is triglyceride (TG) which consists of 50 to 70% particle mass, the remaining 10 to 25% mass consisting of cholesterol ester (CE), and less than 10% of fatty acid [1,2,3,4]. The surface proteins of VLDL include apolipoprotein (Apo) B-100, Apo C-I, Apo C-II, Apo C-III, and Apo E (Figure 1a). Among these, Apo B-100 is the core structural protein and is produced by the liver [5,6]. Apolipoproteins function not only as structural components but also as ligands for cell-surface receptors, and as cofactors for various enzymes such as lipoprotein lipase (LPL) [4,5]. Physiological function, metabolism, and the involvement of disease mechanisms in VLDL are largely regulated and affected by apolipoproteins.

### 1.2. Hepatic Secretion of VLDL

The production and metabolic pathways of VLDL are shown in the Figure 1b. After fat content meal intake, the intestine secretes chylomicron (CM), which reacts with the LPL and transforms into CM remnants. CM remnants are internalized into the liver and are a major resource of triglycerides (TG) and CE. In hepatocytes, TG and CE are transferred to Apo B-100 in the endoplasmic reticulum. The size of VLDL is enlarged with increased TG production in the liver, and the availability of Apo B-100 also affects the lipid content of VLDL [5,7,8,9,10]. After secretion into the blood circulation, VLDL interacts with LPL on the capillary endothelium in tissues (such as adipose, cardiac, and skeletal muscles) where TG is removed from VLDL for storage or utilization [5,10,11,12,13,14]. One study reported that Apo B-48 containing VLDL can be secreted by the intestinal epithelium in animal experiments. However, the physiological role of this alternative VLDL secretion pathway remains unclear [15]. Abnormal VLDL secretion can be related to an imbalance in the intestinal microbiota, and impaired intestinal bacterial flora may also act as a factor in postprandial dyslipidemia [15,16,17,18]. 

### 1.3. Metabolism of VLDL

VLDL can either be hydrolyzed by LPL or taken up by the VLDL receptor (see below). After hydrolysis by LPL, VLDL is transformed into VLDL remnant and intermediate-density lipoprotein (IDL). Apo C-II is transferred to high-density lipoprotein (HDL), which also exchanges TG and phospholipids with CE via the cholesterol ester transfer protein (CEPT). In contrast, VLDL remnants and IDL receive Apo E from HDL [5,14,19,20,21,22]. Approximately half of the IDL can be recognized by the liver using Apo B-100. The rest of the IDL loses Apo E and TG, and becomes low-density lipoprotein (LDL), which is ultimately taken up by the liver via the LDL receptor. Some lipid contents, such as sphingolipids and subclasses of Apo A and C, can affect the metabolism of VLDL [23,24]. For instance, the possession of Apo A-II, A-V, C-II, and C-III makes VLDL associated with insulin resistance [19,25,26,27,28,29]. Further details are provided in the following sections. 

### 1.4. Tissue Expression and Function of VLDL Receptor (VLDLR)

VLDL can be taken up by VLDLR, which is expressed abundantly in adipocytes, cardiomyocytes, and the endothelium [13,30,31]. VLDLR also binds the postprandial remnant-like protein (RLP) in the peripheral tissue. Apo E serves as the ligand for VLDLR; therefore, VLDLR recognizes Apo E-containing lipoproteins, including VLDL, VLDL remnant, and IDL. VLDLR can also bind to molecules such as reelin [13], clusterin [32], and tissue factor inhibitors [33]. VLDLR also interacts with LPL and modulates LDL-mediated TG hydrolysis [34,35,36]. In addition to its important regulatory role in lipid metabolism, VLDLR is found associated with insulin resistance [33,37] and is involved in multiple diseases such as diabetic retinopathy [38], atrial fibrillation [31,39], hypertensive cardiomyopathy [40], and Alzheimer’s disease [13]. 

### 1.5. Physiological Function of VLDL

VLDL production is affected by intake and the secretion is increased during the postprandial state. Excessive nutrition and a high-fat diet (HFD) lead to higher VLDL secretion [41,42], and this phenomenon is more dominant in subjects with hypertriglyceridemia. The postprandial secretion peak of VLDL has been reported with different findings ranging from 30 min to 6 h post-meals. Although there is inconsistency in the timing of peak postprandial VLDL levels, these studies consistently found higher and longer plateaus of postprandial VLDL levels in subjects with metabolic syndrome (MetS) and insulin resistance than in normal subjects [11,39,41,42,43,44,45,46,47]. Other factors that influence VLDL secretion include body status, sex, and race. The mean plasma concentration of VLDL was higher in the obese subjects. Compared with men, women have lower VLDL concentrations and stronger VLDL clearance [48,49]. African Americans secret lesser VLDL than Caucasians [48,50,51]. The cause of the sex difference in VLDL secretion remains unclear, although some studies have shown that sex hormones do not affect VLDL formation and metabolism [52,53]. Except for lipid transportation and metabolism, VLDL can promote thrombin generation and inhibit fibrinolysis [54,55,56], and can bind coagulation factors VII and X [57,58]. Patients lacking Apo B lipoproteins have reduced platelet activation [59]. This clinical phenotype of abetalipoproteinaemia supports the hypothesis that Apo B intervenes in thrombosis regulation. 

## 2. Proposed Pathological Roles for VLDL

### 2.1. Metabolic Associated Fatty Liver Disease (MAFLD) and Hepatitis

The MAFLD subjects have a higher VLDL secretion rate due to increased hydrolysis of intrahepatic TG, and they present with loss of acute reduction in VLDL secretion, but without a difference in Apo B-100 secretion rate [60,61]. The insulin-suppression effect on VLDL is impaired in men with MAFLD regarding the secretion, oxidation, concentration, and reduction of particle size oxidation [62]. In subjects with greater body weight and insulin resistance, Apo C-III of VLDL is increased, which enhances the uptake of VLDL by hepatocytes [20,63,64]. However, in MAFLD patients with severe hepatic fibrosis, plasma TG level, TG ratio of VLDL and circulating total VLDL mass are all reduced [65].

VLDL contributes to sex differences in MAFLD. Men are more likely than women to have MAFLD [66]. For postmenopausal women, the risk for MAFLD is rising due to a decline in estrogen [67,68,69]. A recent animal study showed that estrogen-related receptor (ERR)-α deficiency leads to decreased VLDL secretion, resulting in hepatic lipid accumulation and the development of MAFLD. ERR-α is a nuclear hormone receptor that is involved in multiple metabolic processes [66]. In addition to postmenopausal status, treatment with selective estrogen receptor modulators, such as tamoxifen, suppresses hepatic ERR-α activity and impairs VLDL secretion and promotes hepatic lipid accumulation [66,70]. Therefore, reduced VLDL secretion may be a major contributor to MAFLD in postmenopausal women. Interestingly, the reduced VLDL secretion capability is inheritable as the western diet feeding mother introduces deoxyribonucleic acid (DNA) hypermethylation of hepatic Apo B to male offspring, along with an increased risk of insulin resistance and MAFLD [71,72]. 

Decreased adiponectin levels, which are commonly observed in subjects with MetS and MAFLD, result in increased circulating total VLDL mass and particle amounts [65]. Other molecules involved in MAFLD and VLDL expression include AMP-activated protein kinase (AMPK) and mammalian target of rapamycin (mTOR) [73,74,75]. Ceramide, a type of lipid content in VLDL, is correlated with the severity of MAFLD [76,77]. Plasma dihydroceramides, which are also carried by VLDL, are correlated with MAFLD severity in type 2 diabetes [77,78]. 

Hepatitis C virus (HCV) is capable of binding TG-rich lipoproteins, including VLDL, and forming the lipo-viro-particles (LVPs). For instance, HCV glycoprotein E2 is present in LVPs [79,80], and it protects HCV from antibody-mediated immunoreaction, and promotes virus uptake by lipoprotein receptors in hepatocytes [81]. HCV viral load is negatively correlated with LPL activity, and positively correlated with the Apo C-III content of VLDL [82]. On the other hand, lipid homoeostasis is disturbed by HCV through interaction with the host lipid metabolism via several mechanisms such as increased lipogenesis, reduction of fatty acid oxidation, and reduced TG content in the secreted VLDL [83]. Direct antiviral agents that lead to efficient HCV eradication have been shown to restore the abnormal TG to cholesterol ratios in VLDL, and LDL as well as favorable lipoprotein metabolism [84]. 

Oxidative stress promotes the development of MAFLD in animal study [75]. Nuclear factor erythroid 2-related factor 2 (Nrf2), a transcription factor sensitive to antioxidant responses and activated under oxidative stress, is related to VLDL secretion. Fumarate, an intermediate product of the tricarboxylic acid cycle, is a source of Nrf2 activation. Mice fed an imbalanced diet (high-fat diet or methionine choline-deficient diet) showed decreased Nrf2 expression along with oxidative stress and lipotoxicity of hepatocytes. When the translocation of Nrf2 is suppressed, VLDL maturation is impaired, and lipid accumulation occurs, ultimately leading to hepatic steatosis and MAFLD development [75]. Oxidative stress-induced Nrf2 signaling has also been demonstrated in another animal model of alcoholic hepatitis with elevated VLDLR expression [85]. 

Upregulation of VLDLR expression in the liver, which occurs after prolonged exposure to endoplasmic reticulum stress, occurs via activation of transcription factor 4 (ATF4) signaling and induces hepatic steatosis [86]. Fibroblast growth factor 21 also promotes endoplasmic reticulum stress-related VLDLR overexpression in hepatic steatosis [87]. In addition, peroxisome proliferator-activated receptor (PPAR)-α activation by fenofibrate treatment, which reduces plasma TG levels, is correlated with the modulation of VLDLR expression [88]. 

### 2.2. Insulin Resistance and Metabolic Syndrome

Insulin can suppress VLDL production, and there is increased production and reduced clearance of VLDL in insulin resistance, which is often associated with hypertriglyceridemia [62,89,90,91,92]. Glucagon also reduces hepatic VLDL production [93]. Subjects with insulin resistance have a reduced ability to store lipids in adipose tissue, and are therefore prone to hyperlipidemia and ectopic lipid accumulation [42,94]. In subjects with type 2 diabetes, the ability of insulin to suppress VLDL production is impaired [89,95], and their postprandial VLDL concentration is elevated, although the clearance rate of postprandial VLDL is similar to that in non-diabetic subjects [46]. In studies using ex-labeled VLDL1 and VLDL2 particles to investigate VLDL kinetics during hyperinsulinemia, the size (TG/Apo B ratio) and Apo B-100 concentration were significantly reduced in healthy men but were unaltered in men with type 2 diabetes due to hyperinsulinemia, suggesting that the fatty acid oxidation rate was substantially suppressed in diabetic subjects [89,96]. 

ATP-binding cassette transporter A1 (ABCA1) is an integral cell membrane protein that acts as a mediator of HDL biogenesis. Pancreatic ABCA1 is involved in insulin secretion by β-cells [97]. The ABCA1 of skeletal muscle enhances glucose uptake by enhancing Akt phosphorylation and glucose transporter type 4 transfer to the plasma membrane. Hepatic ABCA1 is involved in the regulation of VLDL production by affecting Apo B trafficking [98]. Patients with complete ABCA1 deficiency had <5% of normal plasma HDL levels and elevated VLDL and TG levels. Genetic variations in ABCA1 are associated with the MetS phenotype, and a broad range of diseases such as coronary heart disease, type 2 diabetes, neurological disorders, and age-related macular degeneration [99]. 

Emerging evidence has shown that in MetS, VLDL becomes cytotoxic, pro-inflammatory, and atherogenic, and is involved in the pathogenesis of various diseases [31,39,100,101,102,103]. Increased Apo C-III, Apo E-III, and E-IV of VLDL in MetS exhibit reduced LPL activity and function [8,27,104,105]. VLDL in MetS exhibits the ability to induce apoptosis by promoting reactive oxygen species production, especially in endothelial cells and sub-endothelial macrophages [59,104]. Furthermore, macrophages stimulated by VLDL lipolysis products exhibit proinflammatory effects [33,106]. The size of VLDL is thought to be related to PPAR-α regulation in the clearance of plasma fatty acids after VLDL hydrolysis [107]. LPL activity affects the diameter of VLDL particles, and large VLDL particles are more abundantly in MetS and insulin resistance [108]. 

### 2.3. Other Endocrinological Disorders

Here we describe several endocrinological disorders and hormones that are correlated with VLDL/TG metabolism. A summary of this section is provided in Table 1.

In Cushing syndrome, increased circulating VLDL and LDL are present as dyslipidemia with elevated plasma CE and TG [109]. Increased VLDL is caused by greater secretion of VLDL than in normal subjects without alteration in VLDL clearance [110]. Exogenous cortisol stimulation with glucocorticoid treatment reduces the degradation of Apo B and increases adipose tissue lipolysis, resulting in elevated circulatory VLDL as well [111]. Increased risk of cardiovascular diseases, and dyslipidemia in Cushing syndrome and in glucocorticoid treatment are all suggested to be correlated with elevated VLDL/TG levels [112,113]. 

The adrenal gland hormone aldosterone is also associated with VLDL. Aldosterone production is promoted by VLDL stimulation and is mediated by the PLC/IP3/PKC signaling pathway [114,115,116]. This mechanism can partially explain how the most commonly used lipid-lowering drug statin is associated with reduced aldosterone levels in hypertensive and diabetic subjects [117]. 

Growth hormone deficiency is related to elevated VLDL production, decreased VLDL clearance, and elevated TG levels. Growth hormone replacement therapy promotes VLDL clearance, but simultaneously increases VLDL secretion from adipose tissue by facilitating lipolysis [110,118]. Therefore, although growth hormones promote fatty acid oxidation, they do not reduce VLDL secretion, but instead can further increase plasma VLDL and TG levels [119]. This phenomenon can explain why subjects with hypopituitarism are prone to cardiovascular and cerebrovascular diseases [120]. The other disease acromegaly with excessive growth hormone has impaired LPL function and increased non-esterified fatty acid in plasma, as well as hepatic overproduction of VLDL [110,121]. 

Thyroid hormones affect LPL activity, cholesterol, and lipoprotein metabolism. In hypothyroidism, LPL function is reduced and the hepatic VLDL secretion rate increases [122,123]. Severe hypothyroidism with thyroid-stimulating hormone (TSH) levels >10 mIU/L is associated with cardiovascular disease and with dyslipidemia, such as elevated total cholesterol, LDL, TG, and lower HDL. Nevertheless, the meta-analysis did not show a significant difference in VLDL, Apo A-I, or Apo B levels [124]. Thyromimetics (thyroid replacement therapy) potentially accelerates energy expenditure and exerts hypolipidemic effects with improved lipid profile [110,125]. 

Regarding sex hormones, testosterone does not affect VLDL metabolism [53,126,127]. While androgen deprivation therapy may elevate VLDL levels [128], testosterone treatment in patients with androgen deficiency has limited effects on VLDL levels [110]. In transgender males, however, increased VLDL/TG is observed after testosterone therapy, probably due to combined suppressed levels of estrogen [129,130,131]. 

In subjects with polycystic ovary syndrome (PCOS), elevated VLDL levels are commonly observed and can improve after effective treatment for PCOS. There is no influence on other lipid profiles [132,133]. The effects of hormone replacement therapy on VLDL metabolism are still controversial. Some studies have shown that estrogen supplementation increases VLDL and body fat mass [134], and similar findings have been observed with progesterone therapy [110,135]. For women of reproductive age, elevated cardiovascular risk is only observed with high-dose estrogen combined with oral contraceptives [136]. Elevated VLDL levels in hormone replacement therapy have also been reported to be associated with increased cardiovascular risk [137]. The impact of sex-affirming hormone therapy on lipoprotein metabolism parameters is largely undetermined. Observational studies have shown the effects of 1 year administration of estrogen or testosterone in transgender persons. The results showed unfavorable changes in the lipid profile in transmen and favorable changes in transwomen [129,131]. 

In pregnant women, VLDL levels increase owing to decreased activities of LPL and hepatic lipase [138,139]. During pregnancy, the production of VLDL is also facilitated by hormone-sensitive lipase in adipose tissue [140]. Fatty acids and cholesterol from VLDL serve as the energy source for the mother and placenta [141]. In pregnant women with gestational diabetes and preeclampsia, VLDL levels also increase significantly due to increased insulin resistance [140,142]. 

Prolactinoma is associated with higher LDL levels; however, the effect of prolactin on VLDL/TG remains unclear [110,143,144,145,146]. The standard therapy for prolactinomas, dopamine agonist therapy, improves insulin sensitivity, decreases LDL-cholesterol levels, and improves the body mass index [124]. The impact of circadian rhythm on disturbing lipid metabolism has been suggested by several animal studies revealing the change in intestinal microbiota as a potential mechanism [147,148,149]. 

**Table 1 ijms-23-04300-t001:** Summary of endocrinological effects on very-low-density lipoprotein (VLDL).

Hormone/Diseases	Effects Related to VLDL	Reference
Cushing syndrome	Increased secretion	[110]
Exogenous cortisol	Declining degradation and increased adipose lipolysis	[111]
Aldosterone	Activation of aldosterone production	[114,115,116]
Growth hormone deficiency	Increased production and decreased clearance	[110,118]
Growth hormone treatment	Increased adipose lipolysis and increased clearance	[110,118,119]
Hypothyroidism	Reduced degradation with increased secretion	[122,123]
Androgen	Androgen-deprivation therapy: increased level	[128]
	Transgender males with testosterone therapy: increased level	[129,130,131]
Polycystic ovary syndrome	Increased level	[132,133]
Estrogen/progesterone therapy	Increased level	[110,134,135]
Prolactinoma	Unclear	[110,143,144,145,146]

### 2.4. Cardiovascular Disorders

The impact of VLDL on cardiovascular diseases has been observed in the correlation between atherosclerosis and coronary events [150]. VLDL is also associated with carotid intima-media thickness and vascular stiffness [151]. Recent studies showed that plasma VLDL level is positively related with major adverse limb events for peripheral arterial occlusive diseases [152,153]. Accumulation of TG-rich lipoproteins contributes to the rupture of atherosclerotic plaques [57]. A large Chinese cohort study showed an association between elevated VLDL concentration and coronary heart disease [154]. CE rather than TG in VLDL particles is correlated with an increased risk of acute coronary syndromes [36]. VLDL levels are also correlated with increased mortality in patients with cardiovascular diseases [155]. Elevated VLDL concentrations increase microvascular events in type 2 diabetes by increasing blood viscosity [156]. Other underlying mechanisms include pro-atherogenic effects, [27,33,37,41,47,157] and promotion of thromboembolism with hypercoagulability [54,56,158]. 

The composition of Apo affects the atherogenic effects of VLDL. Clinical observations have shown that cardiovascular risk is associated with higher Apo B levels and lower Apo C-III of VLDL [3,104,159]. Apo B is suggested to be a major driver in the development of atherosclerosis [8]. In contrast, loss-of-function Apo C-III mutations are associated with low cardiovascular risk [160]. Nevertheless, the effect of Apo C-III on VLDL is complex and it inhibits the interaction between VLDL and its receptors [5]. In addition to Apo B and Apo C, Apo E can also induce atherosclerosis by serving as a ligand for receptors on macrophages, promoting foam cell formation and inflammatory process [161,162,163]. The Apo E of VLDL remnants activates metalloprotease expression and therefore also promotes atherosclerotic plaque rupture [164]. The other pathogenic mechanism of VLDL to atherosclerosis that has been reported by our colleagues is the cytotoxicity exerted by its electronegative charge. The electronegative VLDL has altered lipid content and induces significant apoptosis and senescence of endothelial cells [101,165,166]. 

Beyond atherosclerosis, our team has conducted a series of studies with animal experiments and clinical observations to reveal the pathogenic role of VLDL in atrial remodeling, which is the preclinical stage of atrial fibrillation. VLDL that is isolated from subjects with MetS, rather than VLDL from healthy subjects, induces cytotoxicity and excess lipid accumulation in atrial tissue, disturbs calcium signaling and resulting sarcomere protein derangement, and affects gap junction of cardiac conduction system, leading to atrial remodeling and vulnerability to atrial fibrillation [31,39,100,167]. The electronegatively charged VLDL, particularly postprandial VLDL, may be a significant and independent predictor of atrial remodeling in MetS [39]. 

### 2.5. Neurological Disorders

VLDLR is abundantly expressed in the peripheral nervous system, the cerebellum, and the cerebral cortex. The involvement of VLDLR in Alzheimer’s disease has been observed in clinical observations [168,169,170]. In addition, VLDLR has been detected in senile plaques in the brain [171]. Recent studies have revealed the ability of VLDLR to interact with multiple ligands and molecules such as reelin and clusterin, which are correlated with Alzheimer’s disease [13]. Reelin depletion is regarded as an early phenomenon in Alzheimer’s disease [172,173], whereas clusterin promotes amyloid beta clearance [32,174,175]. Currently, it is suggested that VLDLR is able to interact in the onset and progression of Alzheimer’s disease without the interaction of VLDL. 

There is some evidence suggesting a role for VLDL in psychological disorders. In patients with schizophrenia, data revealed a link to insulin resistance, elevated VLDL levels [176], and increased medium and large VLDL levels [177,178]. Elevated TG/VLDL ratio is also correlated with an increased risk of suicide [179] and impaired cognitive function [180]. In autism spectrum disorder, decreased VLDL levels and Apo B are observed along with increased free fatty acids in the circulation [181]. Sleep quality may affect the VLDL metabolism. An observational study showed that sleep disturbance, sleep medication use, and daytime dysfunction due to poor sleep quality are all associated with elevated serum VLDL levels [182]. This may partially explain the association between poor sleep quality and cardiovascular risk [183]. 

Our colleagues found that VLDL isolated from subjects with MetS was able to induce neuroinflammation and cognitive dysfunction [184]. Typically, VLDL does not pass through the blood-brain barrier (BBB). Studies have suggested that VLDL is enabled by metabolic stress that weakens the BBB and electronegative VLDL can trigger neuroinflammation by activating microglia [102,184]. In addition, increased uptake of VLDL by microglia via the Apo E receptor is associated with impaired LPL function, which affects cognitive function and promotes Alzheimer’s disease [185,186]. Accumulation of VLDL in the medio-basal hypothalamus is also associated with neuroinflammation, regardless of the condition of the BBB [187]. These findings together support the pathogenic role of VLDL in neuroinflammation via microglia and suggest that VLDL is related to an increased risk of cognitive dysfunction in MetS.

### 2.6. Kidney Diseases

Hypertriglyceridemia is the major dyslipidemia in patients with chronic kidney disease (CKD) [188], and the primary cause is the impairment of VLDL clearance [189]. In CKD, VLDL hydrolysis is impaired, and HDL concentration is reduced [26]. Elevated VLDL levels cause oxidative stress in CKD [190]. Moreover, plasma Apo C-III is significantly higher in the CKD group than in normal subjects, which further contributes to insulin resistance and hyperglycemia [191].

VLDL clearance is also reduced in nephrotic syndrome. The mechanism is suggested to be the suppression of VLDLR in an animal study [192], and impaired LPL function due to upregulation of angiopoietin-like protein 4 (ANGPTL4) level [193,194]. In addition, VLDL can be taken up by mesangial cells to exert direct cytotoxicity and cause progression of nephrotic syndrome. Other proposed mechanisms include elevated proprotein convertase subtilisin kexin type 9 (PCSK9), which contributes to elevated LDL and immature HDL levels [195]. 

### 2.7. Inflammation, Autoimmune Disorders and Miscellanies 

Generally, MetS is considered a chronic inflammatory state, and VLDL is positively related to microinflammation in endothelial cells, and to activation of monocytes and expression of cytokines in the extrahepatic tissue [47,100,101,106,196,197]. In insulin resistance and MetS, the expression of ceramides is increased on the plasma membrane [198,199], and the proinflammatory function of macrophages is promoted by the over-absorption of VLDL lipolysis products [77,91,198,200,201,202]. Intracellular ceramides are upregulated in macrophages after incubation with VLDL and exhibit a proinflammatory response [106]. In contrast, multiple cytokines can elevate VLDL levels, such as interleukin (IL)-1, IL-2, and IL-6 [203]. VLDL concentration increases within 2 h after exposure to lipopolysaccharide, and this effect can be sustained for more than 24 h [204]. In an animal study, the proinflammatory effects were more dominant when exposed to postprandial VLDL, which further increased cytokine and integrin activation [45]. Furthermore, a recent molecular study suggested the therapeutic effects of insulin-inducible gene-1 (Insig-1) and gene-2 (insig-2) on excess VLDL biosynthesis and hyperlipidemia. The upregulation of Insig-1 and 2 suppresses the activation of sterol regulatory element-binding protein (SREBP), which is a transcription factor that promotes lipogenesis and is able to inhibit the inflammation induced by VLDL [205]. 

VLDL levels have also been observed in multiple autoimmune disorders. Elevated VLDL levels are observed in patients with systemic lupus erythematosus (SLE) [206,207]. Young women with SLE are at high risk of cardiovascular disease related to dyslipidemia and abnormal VLDL expression [208]. In subjects with primary antiphospholipid syndrome, upregulation of Apo C-III activity leads to decreased VLDL clearance and elevated circulating VLDL levels [209]. In patients with MetS, VLDL levels are positively correlated with disease severity in rheumatoid arthritis [210]. The most commonly used therapy for patients with autoimmune diseases is steroids, which have been proven to increase plasma VLDL concentrations [111,147,148,211]. As VLDL can induce activation of inflammation, monitoring of the lipid profile, including VLDL concentration, is suggested to be important during the treatment and follow-up of patients with autoimmune diseases. 

The effects of VLDL on dermatological diseases have also been demonstrated. In vitiligo, which manifests as pigmentation impairment caused by oxidative stress in melanocytes, elevated TG levels commonly coexisted [212]. In psoriasis, MetS is a poor prognostic factor, and VLDL levels are higher in psoriasis patients with MetS than in those without MetS [213]. In patients with lichen planus, a chronic inflammatory disease affecting the mucosa and skin, significantly elevated VLDL levels and associated cardiovascular risk have also been observed [214]. Whether VLDL is pathogenic in the mechanism of the aforementioned dermatological diseases remains unknown. Nevertheless, elevated VLDL levels can be an important biomarker of cardiovascular risk in these diseased populations.

In the aging population, sarcopenia has been recognized as an important adverse outcome marker and is associated with dyslipidemia, especially VLDL [215]. This observation suggests that skeletal muscle affects the metabolism of VLDL, and vice versa. However, the underlying mechanism remains unclear.

## 3. Association of VLDL in Cancers 

Dyslipidemia is associated with the growth and progression of some malignancies [216]. Similar to ordinary cells, lipids regulate cellular and intracellular signaling, modify the fluidity and lipid rafts of membranes, and directly affect lipid-derived mediators in cancer cells. These mechanisms further affect tumor biology, such as immune escape and cellular invasion [216,217,218]. For instance, high-fat intake has been shown to activate the vascular endothelial growth factor and promote angiogenesis, thereby increasing the proliferation of malignant cells [217]. In highly progressive cancer cells, lipid biosynthesis and uptake were also increased [219,220]. A recent study revealed that CD36, a well-known fatty acid translocase, is correlated with the metastatic ability of cancers and is involved in hepatic VLDL secretion [221]. Upregulation of CD36 is also associated with a higher ability to metastasize, especially for epithelial cancer, as well as promoting VLDL secretion [222]. CD36 is involved in signaling cascades that take up extracellular lipids, and further provides energy for the cell. CD36+ cancer cells are thought to gain the energy for anchoring and surviving at sites distant from the tumor origin [223]. Therefore, increased circulating VLDL levels may be associated with progression of CD36+ malignancies. The interaction between VLDL and malignancies is a novel topic, and in the following section, we summarize recent studies that focus on the relationship between VLDL and the oncogenesis and/or the outcome of cancers.

### 3.1. Breast Cancer

Breast cancer has the most basic and clinical studies focused on the impact of VLDL, and VLDL was found to promote tumor proliferation and progression by promoting angiogenesis, cell migration, and invasion. In a rodent model, breast cancer MDA-MB-231 cells were pre-incubated with charged-defined subfractions of LDL (L1 and L5) and VLDL, and then injected into animals via the tail vein. The results showed that cancer cells incubated with either VLDL, L5, or L1 promoted aggressiveness, but only VLDL incubation exhibited anchorage-independence and caused more lung metastasis. This finding suggests that VLDL promotes lung metastasis in vivo [224]. VLDL uptake brings lipids and offers a sustainable source of energy for cancer cells [225]. The effects of different lipoproteins were compared in human epidermal growth factor receptor 2 (HER2)-overexpressing breast cancer cells and it has been shown that VLDL causes cancer cell growth and morphological changes, and increases cell viability as well [226]. 

Other studies have investigated the role of miRNAs in breast cancer and their relationship to VLDLR expression. The tumor suppressor miRNA-1297 was positively correlated with VLDLR expression. In contrast, oncogenic miRNA-4465 was negatively correlated with VLDLR expression. Reduced miRNA-1297 was associated with decreased VLDLR expression in highly progressive breast cancer. Furthermore, the expression of VLDLR is also negatively correlated with the abundance of Ki-67, a marker of proliferation [227,228]. It is likely that VLDLR expression affects cancer behavior through undetermined mechanisms. In addition, VLDL levels were inversely associated with the development of breast cancer in postmenopausal women [229] and negatively correlated with high mammographic breast density, a risk factor for breast cancer [230]. This paradoxical correlation between VLDL and breast cancer outcomes may be due to the opposing effects of estrogen on VLDL and breast cancer. However, further studies are required to determine the underlying mechanisms. 

### 3.2. Hepatocellular Carcinoma 

Lipid accumulation due to MAFLD and/or genetic predisposition is a risk factor for hepatic cell carcinoma (HCC) [231]. Similar to breast cancer, VLDL also affects HCC development and progression. Recent animal study found that mutation of transmembrane 6 superfamily member 2 (TM6SF2) gene causes worse fibrosis and promotes carcinogenesis of MAFLD [232,233]. TM6SF2 is localized to the membrane of the endoplasmic reticulum, and is essential for the lipidation of Apo B during VLDL synthesis. Impaired TM6SF2 leads to decreased VLDL secretion, which consequently causes hepatic steatosis (MAFLD) and worsens fibrosis in the liver [232]. In hepatoma cells, hypoxia-inducible factor-1 (HIF-1) is upregulated and it can enhance VLDLR expression and VLDL uptake by cells. This mechanism promotes cancer progression [234]. 

### 3.3. Other Cancers 

VLDL levels were elevated in lung cancer patients compared to non-cancer subjects [235]. However, in the lung cancer population, HDL level was the only prognostic factor [236]. Nevertheless, the role of VLDL in lung cancer remains unclear. 

Two Indian clinical studies revealed an inverse correlation between serum VLDL levels and leukoplakia, which is a precursor lesion of oral cancer [237,238]. The major risk factors for oral cancer are smoking, alcohol consumption, and betel nut consumption. These factors increase free-radical production and further damage the cell membrane. Decreased VLDL levels are suggested to be a consequence of increased consumption due to cell membrane repair and oxidative stress [239]. Serum VLDL level has also been suggested as an indicator of the severity of oral cancer and pre-cancer lesions. 

While studies focusing on VLDL and other cancers are still sparse, numerous studies have shown evidence of a correlation between elevated TG levels and carcinogenesis risk for small cell lung cancer, breast cancer, pancreatic cancer, and ovarian cancer [217,240]. However, the underlying mechanisms remain largely unknown.

## 4. Conclusions

VLDL is produced in the liver and is involved in MAFLD, MetS, atherosclerotic diseases, cognitive dysfunction, autoimmune disorders, HCC and breast cancers. Moreover, VLDL inherits cytotoxicity from MetS with undetermined mechanisms and may cause atrial myopathy in the pre-clinical stage of atrial fibrillation. VLDL levels are affected by multiple endocrine systems and VLDL promotes aldosterone secretion. The proposed pathological roles of VLDL awaiting determination include the extrahepatic secretion of VLDL, sleep disorders, neurodegenerative diseases, and a large spectrum of cancers. Overall, there is emerging evidence supporting the pathological roles of VLDL in various diseases across multiple systems. Elucidation of the quality control and metabolism of VLDL, instead of the sole correlation of the VLDL level, may enhance our understanding of its contribution to health and disease mechanisms. Ultimately, the potential VLDL-targeted therapies cab be discovered and be successful. 

## Figures and Tables

**Figure 1 ijms-23-04300-f001:**
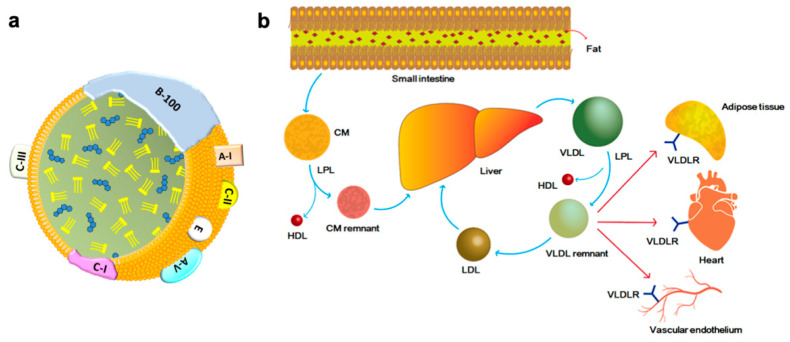
Structure (**a**) and metabolism (**b**) of very-low-density lipoprotein (VLDL). VLDL is a triglyceride (TG)-rich lipoprotein composed of TG, cholesterol ester (CE), phospholipid membrane, apolipoprotein (Apo) A, B-100, and C. The TG consists of 50–70% total mass, and CE consists 10–25%. The remainings are proteins and fatty acids. After fat-content meal intake, chylomicron (CM) is secreted by the intestine into blood circulation, and later transformed to CM remnant by lipoprotein lipase (LPL) and ultimately internalized into the liver where VLDL is produced. In circulation, the hepatic secreted VLDL is transformed to VLDL remnant by the activity of LPL. Both VLDL and VLDL remnant bind VLDL receptor (VLDLR), which is expressed in adipocyte, cardiomyocyte, vascular endothelium in various tissues.

## Data Availability

Not applicable.

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
