# Peer review of "Emerging Evidence of Pathological Roles of Very-Low-Density Lipoprotein (VLDL)"

_ijms, 2022, doi:10.3390/ijms23084300_

Round 1

Reviewer 1 Report

In the present manuscript, the authors presented a review of the pathological roles of very-low-density lipoprotein (VLDL).

The authors made an excellent minireview regarding the physiological function of VLDL as well as its implication in numerous pathological states, such as fatty liver disease, hepatitis, insulin resistance, metabolic syndrome, cancer, endocrinological, cardiovascular, neurological, kidney and autoimmune disorders. Unfortunately, in the case of the majority of these disorders, the underlying mechanisms remain largely unknown.

It is obvious that the pathological roles of VLDL can influence the quality of human life, therefore this manuscript could be useful for discovery of potential therapeutic targets. Thus, it would be interesting if the authors could present several VLDL-based targets, already known in the literature.

Minor text revisions are required before publication. For instance, the phrase written at row 32 seems to be unfinished and should be completed. All the abbreviation should be mentioned in the Abbreviations Section.

Author Response

Point 1:  It is obvious that the pathological roles of VLDL can influence the quality of human life, therefore this manuscript could be useful for discovery of potential therapeutic targets. Thus, it would be interesting if the authors could present several VLDL-based targets, already known in the literature.

Response 1: 

We thank the reviewer for pointing out the significance of this review and all your comments. 

Regarding the application of VLDL as a therapeutic target, there is no available publication in the literature search. In the revised manuscript, the following statement has been added in the Conclusion: “Ultimately, the potential VLDL-targeted therapies can be discovered and be successful.” (Row #490–491, Page 11 of 20).

Point 2: Minor text revisions are required before publication. For instance, the phrase written at row 32 seems to be unfinished and should be completed. All the abbreviation should be mentioned in the Abbreviations Section.

Response 2:

The last two sentences of the abstract have been amended (Row #31–34, Page 1 of 20). The Abbreviation Section has been added (Row #508–547, Page 11–12 of 20).

Reviewer 2 Report

This is a very interesting paper focused on relevant topic, well written and very informative. In my opition this good paper is suitable of pubblication on this special issue. 

Author Response

Point 1:  This is a very interesting paper focused on relevant topic, well written and very informative. In my option this good paper is suitable of publication on this special issue. 

Response 1:

Thank you for your favorable comment. 

Point 2: Moderate English changes required

Response 2: Moderate English changes is done.

Reviewer 3 Report

Dear Authors, 

You have a very useful review. I have only two tiny questions.

In my opinion, a short abstract about the changing and role of VLDL during pregnancy might be interesting. Also, you can add more data in the context of oncological diseases or oncology in general. For me - not enough for now.

Author Response

Point 1: You have a very useful review. I have only two tiny questions. In my opinion, a short abstract about the changing and role of VLDL during pregnancy might be interesting. 

Response 1: Thank you for your favorable and helpful comments. The regulation of VLDL is important during pregnancy and it can be another informative mini-review. Briefly, we added statement regarding the changing and roles of VLDL during pregnancy in row# 262–267 (Page 6 of 20).

Point 2: Also, you can add more data in the context of oncological diseases or oncology in general. For me - not enough for now.

Response 2: 

The context of oncology in general has also been amended with more data (row# 412–423, Page 9 of 20).